# Study on the Coupling Effect of Agricultural Production, Road Construction, and Ecology: The Case for Cambodia

**Lingfei Weng \*, Wentao Dou and Yejing Chen**

School of Public Policy and Administration, Chongqing University, Chongqing 400044, China
\* Correspondence: wenglf@cqu.edu.cn

**Abstract:** Agricultural development is a necessary component of national development efforts to fight food crises and promote poverty reduction in many developing countries. However, many developing countries have fallen into a stalemate between modernization and development—modernized areas are less capable of driving regional development despite their abundant land and rich population. Striking a balance between agricultural technology and environmental protection is a key feature of sustainable land development. Based on the social–ecological resilience theory, this study takes Cambodia as an example and aims to establish a comprehensive evaluation index system to measure the agricultural production, regional road construction, and agricultural eco-environment in Cambodia. The coupled coordination model and gray relation analysis model are utilized to explore the interaction between agriculture, roads, and the agricultural eco-environment. The results show that (1) Cambodia has road environmental risks, and there is a need for rural labor migration in areas with higher levels of economic development. (2) The main agricultural production areas are faced with the dilemma of lagging infrastructure development, such as roads, and a huge potential for agricultural development. (3) In the plains areas, the growing population has caused tension between food security, fertilizer abuse, and deforestation, which intensified the disturbance of the agricultural ecological environment. In summary, based on their own developmental needs, developing countries at different stages of development can explore the interaction between agricultural production, infrastructure development, and the agricultural eco-environment in the process of agricultural development. This study attempts to provide a set of practical development policy implications for developing countries that are seeking to enhance the coupling relationship between agricultural production, infrastructure, and the agricultural eco-environment.

**Keywords:** agricultural production; road infrastructures; agricultural eco-environment; coupling analysis



## 1. Introduction

Agriculture provides the necessary material basis for human survival and development and is the basic industry that supports the national economy of Cambodia. Under the negative influence of COVID-19, global food security was compromised, and the number of people who went hungry globally maintained an upward trend in 2020 [1]. At present, due to the influence of the Russia–Ukraine conflict, food production has suffered another setback, leading to increasing global food trade costs. It's expected that the global demand for food production will double by 2050 [2], which undoubtedly poses a serious challenge to global agricultural production.

The agricultural production capacity of human beings has significantly improved over time. Improvements in agricultural technologies have led to improved agricultural productivity. The world is witnessing an increasing input of agricultural production factors such as natural resources, labor, capital, and agricultural science and technology. For developing countries, not only can agricultural development become the engine of economic growth, but it can also help avoid major economic risks and maintain social stability [3].

However, many developing countries fall into a paradox. They are endowed with a large population and rich natural resources but have an inadequate level of modernization. Historically, the globalization of agricultural crops allowed for the feeding of more people. In many developing countries, the lack of internal impetus for modernization, corruption, and limited governance further prevent such countries from taking advantage of globalization while addressing domestic social issues, such as poverty and education.

By relying solely on foreign investment as the external impetus, many resources are allocated to the areas that possess locational advantages in these countries, such as in terms of political, transportation, and natural resources. However, these areas, often with pre-modern infrastructure and institutions, have not yet helped to lift these nations out of poverty. Many people in such areas have poor living standards, are affected by the risks brought by modern society, such as climate change and pandemics, and are unable to enjoy the benefits of modern institutions. Therefore, to solve the global food crisis, eliminate poverty, and reduce inequality across regions, regulating resource misallocation to drive domestic production is key, which will thus enhance returns in order to achieve the development of agriculture.

As a new industrial system and developmental stage of agriculture, modern agriculture's ultimate purpose is to achieve a rough convergence of the return on investment in agriculture with that of other industries [4]. Therefore, it is of great significance to undertake the following: (1) improve agricultural land systems, production, science, and technology; (2) realize the linkage between the agricultural sector and other sectors; (3) highly integrate the three industries of manufacturing, service, and agriculture; (4) establish a market-oriented agricultural commercialization and industrialization management model [5]; and (5) improve the agricultural operators' human capital level, thus enhancing the modern attributes of the input factors and reducing their transaction costs.

Road infrastructure is not only a necessary condition for social productivity and livelihoods but also the carrier of regional economic operation. Road infrastructure expands the market scope and realizes economies of scale and specialized production by reducing transaction costs in the spatial dimension, which in turn promotes competition and the diffusion of knowledge and technology [6,7]. In developing countries, not only can the improvement of road infrastructure conditions increase the opportunities for the rural labor force to transfer to non-agricultural industries, but it also reduces the mismatch of social factors of production in various fields [8]. Meanwhile, rural road facilities can reduce the spatial distance between farmers and the market, the time for agricultural products to reach markets, and transaction costs, such as labor and transportation. They can also motivate farmers to increase commodity inputs, promote specialized production, achieve economies of scale, promote the development of the non-agricultural sector in rural areas, thus increasing agricultural profitability, and play a central role in rural development [9].

With the help of the regional road system, agricultural production is capable of positively interacting with other social production activities. A steady transfer of rural labor is created when the yield rate of non-agricultural work exceeds the yield rate of agriculture. In this process, on the one hand, urban production obtains the required labor capital and starts to expand and increase in value. On the other hand, in rural production, the labor input factors are replaced by other modern factors. Modern systems such as the market, property rights, and credit are popularized in rural areas. Social production factors are effectively allocated, and the regional economy grows. When there is a socio-economic crisis and the yield rate of non-agricultural work is lower than the yield rate of agriculture, a reasonably modern system contributes by providing convenience and support for the return of rural workers to ensure employment.

However, the ecosystem is an important constraint for both agricultural production and road infrastructure development. Agro-environment refers to the quantity and quality of agricultural land, water, and biological and climatic resources that support human survival and development [10]. Thus, a good agro-ecological environment is a key component of sustainable rural socio-economic development, upon which agricultural production

is highly dependent [11,12]. While the transition to the modernization of agricultural production has increased the agricultural production efficiency, there is a risk of ecosystem neglect, leading to a series of agro-ecological problems, such as soil degradation, water eutrophication, soil erosion, fragmentation of agricultural landscapes, agricultural non-point source pollution, loss of ecological balance, and a reduction in species diversity. Although ecosystems have a certain self-healing capacity for external disturbances, i.e., socio-ecological resilience [13], when the disturbance exceeds the resilience, the system will enter an unpredictable state, the result of which is often beyond human comprehension. In many developing countries, the deterioration of the agricultural eco-environment (AEE) is becoming increasingly prominent, leading to rural pollution, and is seriously threatening the sustainable development of food production and agriculture, thus further decreasing the livable land area for human beings.

Road facilities, as man-made landscapes superimposed on ecological landscapes, generally negatively impact ecosystems on three different levels: climate, species, and ecological landscapes. This is evidenced by the impacts such as air and water pollution, increased temperatures at road areas, changes in population densities, a reduction in species numbers or extinction of species, reduced biodiversity, and the fragmentation of natural habitats [14]. Even though the benefits of road infrastructure are many, expanding this infrastructure type should be done with caution.

Previous studies have provided useful references for qualitative and quantitative studies related to the relationship between agricultural production, regional road construction, and the agro-ecological environment. It is necessary to build regional road systems for agricultural production under environmentally acceptable conditions to help developing countries improve their agricultural systems in a smooth manner. For most developing countries, rural areas are still the most important production and settlement areas [15], and the food production issue is pressing. Previously some studies have been conducted on the interaction between the agro-environment, agricultural production, and road construction, however there is a lack of integrated approach that determine a successful interaction among three of them. In large part, this is due to the fact that the policies that focus on these three factors do not tend to occur simultaneously in more developed countries. Only developing countries are likely to make these fields a top priority within their national development strategies, hoping to achieve breakthroughs in these fields by taking advantage of the latecomers and catching up to the leaders. However, it is hard to hear such countries asking for help in the highly competitive international arena.

Agriculture plays a pivotal role in Cambodia's national economy. Despite the constraints, such as lagging infrastructure and technology and lack of financial and human capital, Cambodia is rich with an ample labor force, great market potential, and substantial agricultural resources, such as rice, soybeans, corn, cassava, and cashew nuts. The Cambodian government has made agriculture a priority on their national development agenda, with approx. 85% of the population engaged in agriculture across approx. 6.7 million hectares of arable land in the country. The exports of agricultural products account for a large proportion of the total export value, and the government has made every effort to improve agricultural production and its investment environment. As a lower middle-income country, Cambodia is an ideal example of the interaction between agricultural production, regional road construction, and the agricultural eco-environment. With a relatively stable political society, open policies, and rich natural resources, Cambodia is one of the most attractive destinations for investment among the ten countries that have joined the Association of Southeast Asian Nations (ASEAN). However, although financing for infrastructure development takes up a considerable share of investment, the current transportation infrastructure development is unable to meet the country's economic development needs [16,17]. Currently, Cambodia is promoting the "Rectangular Strategy", focusing on the renewal and reconstruction of its infrastructure. What's more, according to the different levels of local and regional development in Cambodia, it is possible to yield twice the result with half the effort by rationalizing the industrial policies and utilizing resources efficiently.

As a result, given the above considerations, this study comprehensively evaluates the level of agricultural production, regional road construction, and the agricultural eco-environment in Cambodia in 2019. It then explores the interaction between agricultural production, regional road construction, and the agricultural eco-environment using the coupling coordination model, and finally analyzes the impact of regional road construction and the agricultural eco-environment on agricultural production using the gray correlation model.

## 2. Overview and Data Source

### 2.1. Overview

The Kingdom of Cambodia, abbreviated as Cambodia (102°18′–107°37′ E, 10°20′–14°32′ N), has 25 provincial (municipal) administrative divisions. In 2022, it had a total population of roughly 16.9 million [18]. It is located in the China–Indochina Peninsula, bordering Thailand, Laos, and Vietnam. As one of the ASEAN member countries, Cambodia's current economy is dominated by agriculture, with a strong cultural tourism industry and a weak industrial base.

Cambodia has a tropical monsoon climate with extensive forest cover and many islands along the coast. The rainy season is from May to October, and the dry season is from November to April. Affected by the terrain and monsoons, the annual average temperature is 29–30 °C, and the precipitation varies greatly from place to place [19]. The eastern, western, and northern parts of the country are surrounded by mountains and plateaus, while the central and southern parts are located on plains. The shape of Cambodia is similar to a dustpan, with a land area of 181,035 square kilometers.

According to its geographical features, Cambodia can be divided into four parts: plains, Tonle Sap Lake, plateau mountains, and coastal areas. The capital, Phnom Penh, is located in the southern plains, as shown in Figure 1. As the largest lake in the China–Indochina Peninsula, Tonle Sap Lake covers an area of more than 2500 square kilometers at low water levels and 10,000 square kilometers in the rainy season, which has a regulating effect on the Mekong River flood [19]. The Mekong River and Tonle Sap Lake are connected by the Tonle Sap River, thus effectively regulating the local ecological environment to safeguard the agricultural development of Cambodia.

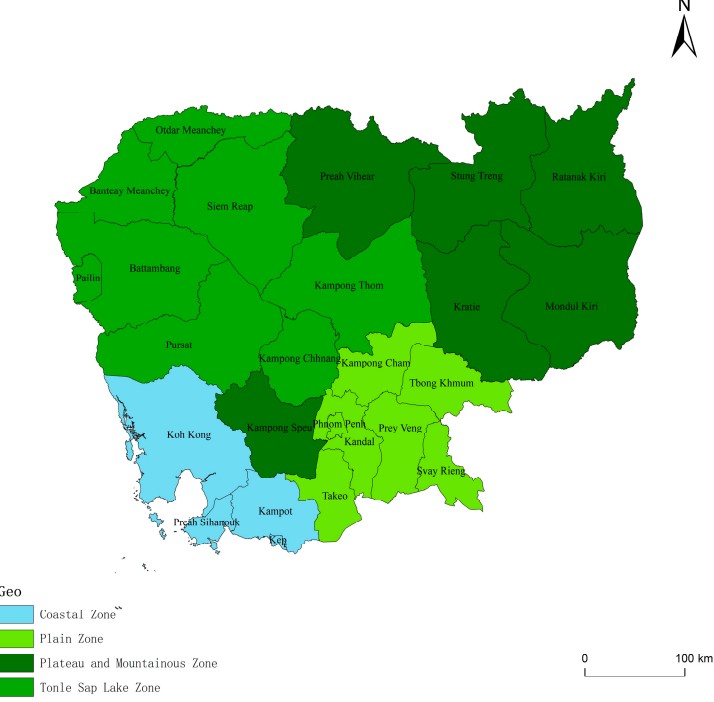

**Figure 1.** Geographical map of Cambodia.

## 2.2. Data Source

The administrative division data of Cambodia were from the Database of Global Administrative Areas (GADM) (http://www.gadm.org/, accessed on 15 March 2022.), the road network data were from the OSM (https://www.openhistoricalmap.org/, accessed on 15 March 2022), and the agricultural and population data of Cambodian provinces (cities) in 2019 were comprehensively taken from the "Agricultural survey between two censuses in Cambodia 2019", and the "General Population Census of Cambodia 2019" (http://www.nis.gov.kh/, accessed on 15 March 2022). The enhanced vegetation index (EVI) and the net primary production (NPP) of Cambodia in 2019 were taken from the LPDAAC (https://lpdaac.usgs.gov//, accessed on 22 March 2022), with a spatial resolution of 1 km and 500 m. The Cambodian 2019 land cover map for calculating the relevant landscape index was from the LPDAAC (https://lpdaac.usgs.gov//, accessed on 22 March 2022), with a spatial resolution of 500 m. The landscape classification standard refers to the classification method of the University of Maryland; the Cambodian land cover for calculating the cultivated land area image was from Globalland30, with a spatial resolution of 30 m (http://www.globallandcover.com/, accessed on 9 April 2022).

The raw data were standardized by the maximum–minimum difference normalization method regarding the previous studies, and the results are as follows [20–22].

$$x'_{i,j} = \begin{cases} \dfrac{x_{i,j} - \min(x_j)}{\max(x_j) - \min(x_j)}, & \text{Positive indicator} \\[2ex] \dfrac{\max(x_j) - x_{i,j}}{\max(x_j) - \min(x_j)}, & \text{Negative indicator} \end{cases} \tag{1}$$

In the formula, $x_{i,j}$ is the jth indicator of the ith province (city), $x'$ is the dimensionless data of $x$, and $\max(x_j)$ and $\min(x_j)$ are the maximum and minimum values of the jth indicator in all the provinces (cities).

## 3. Methods

### 3.1. Research Design and Framework

The coupling relationship between agricultural production, regional road construction, and the agro-environment is an open and complex system involving nature, the economy, and society. We tried to establish their interactive relationship, as shown in Figure 2. Agricultural production, based on natural inputs such as animals and plants (or plant seeds) and natural conditions such as heat, light, water, terrain, and soil, refers to the outputs after inputting labor. Utilizing developed agricultural technology such as agricultural machinery, fertilizers, pesticides, and breeding seeds can promote the comprehensive output of agriculture and reduce the labor input in agricultural production while imposing a certain burden on the ecological environment. In a broad sense, road infrastructure is the material project that provides public services for social production and residential livelihood. In this study, it refers to the regional transportation infrastructure dominated by highways. On the one hand, facilities such as roads and highways promote or restrict the transition of traditional agricultural production. On the other hand, the enhanced agricultural production capacity has higher requirements for road carrying capacity, information flow, and economic flow, which in turn contributes to the improvement of road infrastructure. Therefore, the coordinated development between them is not only the objective requirement of social efficiency but also the constraint of reality.

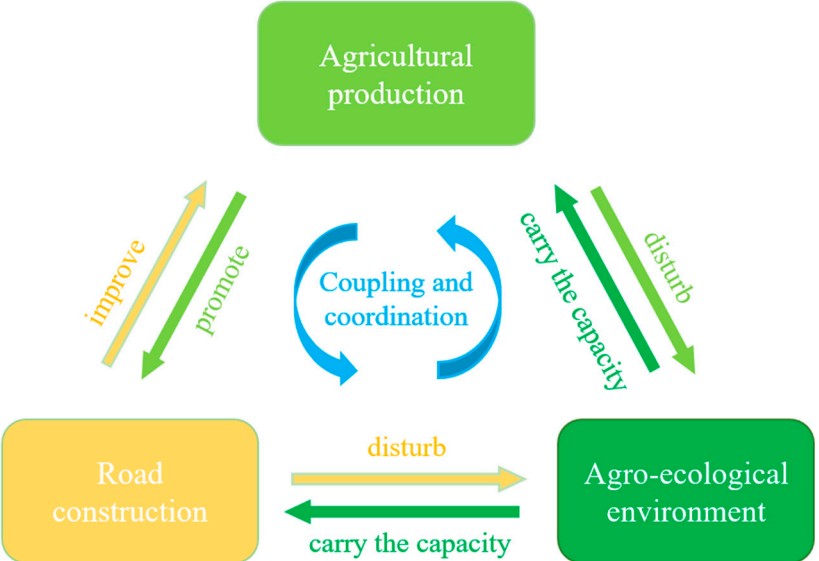

**Figure 2.** Research framework.

Both agricultural production and road infrastructure have certain negative impacts on ecosystems. Once the negative impacts exceed what the ecosystem can bear, it brings unpredictable consequences, further compressing the living space of human beings and causing a detriment to the sustainable development of the local area. It is thus a rational choice to coordinate between the regional development and the corresponding ecological environment.

Complex interactive and coupled links exist between agricultural production, regional road construction, and the agricultural eco-environment, and they are mutually constrained, supported, and related. Many developing countries are faced with the development dilemma for diverse and complex modernization needs and limited policies that can be implemented. An in-depth exploration into the interaction between the three can achieve many things, including reducing transaction costs, improving agricultural production efficiency, ensuring food security and sustainable agricultural development, realizing the transition from traditional to modern agriculture, and eliminating regional poverty and inequality in developing countries by improving road infrastructure in an environmentally controlled manner according to the needs of agricultural production.

### 3.2. Measurement of Coupled Subsystems

To explore the coupling relationship between agricultural production, regional road construction, and the agricultural eco-environment, this study established a comprehensive evaluation index system based on the previous research (Table 1) and following the principles of the scientific and simplistic selection of indicators [23]. In the evaluation system, the three subsystems were determined by their respective indicators.

The agricultural production systems were measured by both the input and output sides. The output side included the output per capita, the proportion of the commercialized agricultural products, and the ratio of the harvested area to planted area, which reflected the annual production efficiency. For the input side, this study focused on the human capital in agricultural production, aiming to alleviate the mismatch of labor factors that often occurs in developing countries [24]. Therefore, the indicators of agricultural production included the proportion of rural residents in the whole society, the proportion of agricultural employees in the agricultural population, the annual agricultural labor hours which reflect the physical input of labor, and the education level which indicates the intellectual input. Other agricultural input factors, such as land, fertilizers, agricultural facilities, and so on, were analyzed as the panarchy pressure on the ecosystem [25,26].

**Table 1.** Agricultural production, regional road construction, and agricultural eco-environment evaluation index system.

| System | Subsystems | Indicators | Unit | Weights | Direction |
|---|---|---|---|---|---|
| Agricultural production | Outputs | Per capita output value of agriculture, livestock, and fisheries | Riel/person | 0.4165 | + |
| | | Proportion of commercialized agricultural products [27] | % | 0.1304 | + |
| | | Proportion of harvest | % | 0.1082 | + |
| | Inputs | Annual time spent in agricultural production [28] | h | 0.0659 | − |
| | | Education level of rural population [29] | Year/person | 0.0581 | + |
| | | Proportion of rural residents of the greater population | % | 0.1374 | − |
| | | Proportion of rural residents–agricultural employees in the rural population | % | 0.0834 | − |
| Road facilities | Supply level | Road density | km/km$^2$ | 0.4719 | + |
| | | Road grade [23] | | 0.0826 | + |
| | | Road connectivity [23] | | 0.1272 | + |
| | Service level | Accessibility [30] | h | 0.0644 | − |
| | | Technology level [31] | person/km | 0.2524 | + |
| Agro-environment | Structure | Shannon's diversity index (SHDI) [32] | | 0.1119 | + |
| | Self-cleaning capacity | The enhanced vegetation index (EVI) [33] | | 0.0897 | + |
| | | Net primary production (NPP) [34] | | 0.3080 | + |
| | Panarchy pressure | Patch density (PD) [35] | | 0.1477 | − |
| | | Arable land per capita | km$^2$/person | 0.1703 | + |
| | | Fertilizer usage per hectare | Riel/km$^2$ | 0.0983 | − |
| | | Multiple cropping index | % | 0.0741 | − |

Note: + and − represent both positive and negative indicators respectively.

The road infrastructure was measured by the level of supply and service. The supply level included the area density of the road, the level of the road classes, and the road connectivity. The service level included the road accessibility, and the level of technology that reflects the use of different roads by the residents. Based on the level of ecosystem resilience and information from the previous related studies, seven representative indicators were selected from three aspects to evaluate the level of the agro-ecological environment, including the agro-ecosystem structure, ecosystem self-cleaning capacity, and ecosystem panarchy pressure. Shannon's diversity index (SHDI), the enhanced vegetation index (EVI), and the net primary production (NPP), excluding the man-made landscape, were selected to assess the level of the ecosystem structure and self-purification capacity of the agro-ecosystem. Four other variables—the patch density (PD), arable land per capita, fertilizer usage per hectare, and the proportion of crop sown area to arable land area, i.e., multiple cropping index—were selected to evaluate the ecosystem panarchy pressure.

In order to reduce the influence of the subjective factors in the evaluation process, this study adopted the entropy method to determine the weights of the indicators in the evaluation system. A comprehensive evaluation of each system was obtained using the following formula:

$$U_i = \sum_{j=1}^{k} w_j \times x'_{i,j} \tag{2}$$

where $U_i$ is the overall evaluation of the subsystems (agricultural production, regional road construction, and the agro-ecological environment) in each province (city), $W_j$ is the weight of indicator $j$, and $x'$ is the standard value of evaluation indicator $j$.

### 3.3. Coupling Analysis of Agriculture, Roads, and the Agro-Environment

The coupling degree is often used to measure the degree of coupling and coordinated development of integrated systems, such as institutions or socio-economic systems [36]. The specific formula is:

$$
\begin{aligned}
C &= K \times \left\{ (U_1 \times U_2 \times \ldots \times U_k) / \left[ \prod_{1 \leqslant i, i \notin k, i+j} U_i + U_j \right] \right\}^{1/k} \\
T &= aU_i + bU_j + cU_k (i \neq j \neq k) \\
D &= (C \times T)^{1/2}
\end{aligned}
\tag{3}
$$

where $U_i$ represents the integrated value of system $i$, and $C$ denotes the coupling degree of $k$ systems (ranging from 0 to 1), which reflects the degree of interaction between system $i$ and system $j$. According to the changed characteristics of the coupling degree, the coupling degree is divided into four different stages: the low coupling stage (0.0–0.3), antagonistic stage (0.3–0.6), grinding stage (0.6–0.8), and high coupling stage (0.8–1.0). $D$ indicates that the coupling coordination degree between the systems ranges from 0 to 1. The higher $D$ is, the higher the level of consistency between the subsystems. $T$ represents the overall level of the three systems, and $a$, $b$, and $c$ denote the undetermined coefficients.

In this study, agricultural production, regional road construction, and the agro-environment were considered equally important, so their values were each set as one-third. According to a related study, the coupling degree between these two subsystems was divided into two categories and five subcategories, as shown in Table 2 [37].

**Table 2.** Coupling coordination level.

| Grade (D) | Type |
| --- | --- |
| 0 < D ≤ 0.2 | Severe incongruity |
| 0.2 < D ≤ 0.4 | Incongruity |
| 0.4 < D ≤ 0.6 | Low-level coordination |
| 0.6 < D ≤ 0.8 | Basic coordination |
| 0.8 < D ≤ 1.0 | High-level coordination |

The coupling relationship between the subsystems was considered in two steps. Firstly, depending on whether agricultural production or road facilities were being analyzed, the comprehensive level ($U_{AP}$, $U_{RE}$) should have been within the bearable range of the agro-ecosystems ($U_{AEE}$). On the other hand, the agro-ecosystems should have been fully utilized by agricultural production when the conditions were suitable. Secondly, there should not be a large gap between $U_{AP}$ and $U_{RE}$. Therefore, the coupling relationship between subsystems was further evaluated according to the magnitude of the difference between the integrated levels of the subsystems. In line with the previous studies, it was considered that there was no gap when $|U_i - U_j|$ was not greater than 0.1; there was a gap when it was greater than 0.1 but less than 0.2 [38]; and there was a serious gap when it was greater than 0.2, as classified in Table 3.

**Table 3.** Coupling relationship between the subsystems.

| Type of Interaction | Relationships | Subcategories |
|---|---|---|
| AEE-AP | $0 \leq U_{AEE} - U_{AP} < 0.1$<br>$-0.1 \leq U_{AEE} - U_{AP} < 0$<br>$-0.2 \leq U_{AEE} - U_{AP} < -0.1$<br>$U_{AEE} - U_{AP} < -0.2$<br>$0.1 \leq U_{AEE} - U_{AP} < 0.2$<br>$U_{AEE} - U_{AP} > 0.2$ | Acceptable<br>Low risk<br>Risk exists<br>High risk<br>Potential<br>High potential |
| AEE-RE | $U_{AEE} - U_{RE} > 0$<br>$-0.1 \leq U_{AEE} - U_{RE} < 0$<br>$-0.2 \leq U_{AEE} - U_{RE} < -0.1$<br>$U_{AEE} - U_{RE} < -0.2$ | Acceptable<br>Low risk<br>Risk exists<br>High risk |
| AP-RE | $\mid U_{AP} - U_{RE} \mid \leq 0.1$<br>$-0.2 \leq U_{AP} - U_{RE} < -0.1$<br>$U_{AP} - U_{RE} < -0.2$<br>$0.1 < U_{AP} - U_{RE} \leq 0.2$<br>$U_{AP} - U_{RE} > 0.2$ | No lag behind<br>Agricultural demand exists<br>High agricultural demand<br>lag behind<br>Severe lag behind |

Note: AP, RE and AEE respectively represent agricultural production system, road environment and agro-ecological environment;.

### 3.4. Entropy-Weighted Gray Correlation Model

To further explore the relationship between agricultural production, road infrastructure, and the agricultural eco-environment, an entropy-weighted gray correlation model [39], which does not require a strict sample size, was used to analyze the influencing factors.

The agricultural production variable was set as the reference sequence $X_0$, $X_0 = \{X_0(1), X_0(2), \ldots, X_0(n)\}$, and the road and agricultural eco-environment variables were set as the comparison sequence $X_i$, $X_i = \{X_i(1), X_i(2), \ldots, X_i(m)\}$. The equations used are as follows:

$$\xi(x_0(k), x_i(k)) = \frac{\min_i \min_k \left| x_0'(k) - x_i'(k) \right| + \rho \max_i \max_k \left| x_0'(k) - x_i'(k) \right|}{\left| x_0'(k) - x_i'(k) \right| + \rho \max_i \max_k \left| x_0'(k) - x_i'(k) \right|} \tag{4}$$

$$\gamma(X_0, X_i) = \frac{1}{s} \sum_{j=1}^{s} \xi(x_0(k), x_i(k)) \tag{5}$$

$$R = \sum_{j=1}^{n} \omega_j \gamma(X_0, X_i) \tag{6}$$

In the above formula, $\min_i \min_k \left| x_0'(k) - x_i'(k) \right|$ is the two-level minimal difference, $\max_i \max_k \left| x_0'(k) - x_i'(k) \right|$ is the two-level maximum difference, $\xi(x_0(k), x_i(k))$ is the correlation coefficient between the $k$th indicator of the comparison sequence $X_i$ and the $k$th indicator of the reference sequence $X_0$, $\rho$ is the resolution coefficient, and the interval is $(0, 1)$, usually taken as 0.5. $\gamma(X_0, X_i)$ is the correlation between the $k$th indicator of the sequence $X_i$ and the kth indicator of the reference sequence $X_0$, s is the sample size included in the model, and R denotes the overall correlation degree. The same entropy method was used to determine the indicator weights, as shown in Table 4.

**Table 4.** Entropy weight grey correlation model variables.

| System | | Variable | Weight |
|---|---|---|---|
| Reference sequence $X_0$ | | Per capita output value of agriculture, livestock, and fisheries | 0.4165 |
| | | Proportion of commercialized agricultural products | 0.1304 |
| | | Proportion of harvest | 0.1082 |
| | | Annual time spent in agricultural production | 0.0659 |
| | | Education level of rural population | 0.0581 |
| | | Proportion of rural residents of the greater population | 0.1374 |
| | | Proportion of rural residents–agricultural employees in the rural population | 0.0834 |
| Comparison sequence $X_i$ | Road facilities | Road density | |
| | | Road grade | |
| | | Road connectivity | |
| | | Accessibility | |
| | | Technology level | |
| | Agri-ecosystem | Shannon's diversity index (SHDI) | |
| | | The enhanced vegetation index (EVI) | |
| | | Net primary production (NPP) | |
| | | Patch density (PD) | |
| | | Arable land per capita | |
| | | Fertilizer usage per hectare | |
| | | Multiple cropping index | |

## 4. Results Analysis

*4.1. Spatial Pattern of Agricultural Production, Road Construction, and the Agricultural Eco-Environment*

Agricultural development in Cambodia is still dominated by self-sufficient small-holders. This is due to a lack of roads and other associated infrastructures. Agricultural technology is underdeveloped, and farmland is highly fragmented and extensive. The overall domestic evaluation of agricultural production (AP) in Cambodia was significantly lower than that of the agro-ecological environment (AEE) and higher than that of road infrastructure (RE), as shown in Table 5. The composite level of the comprehensive evaluation of Cambodian agricultural production was low, and there was no spatial autocorrelation. The first grade of the comprehensive evaluation is Oddar Meanchey, upstream of Tonle Sap Lake; Kampong Thom, downstream; and Stung Treng, in the highland mountainous area. However, the Tonle Sap Lake area and the southern plain with superior natural conditions have yet to be well developed.

**Table 5.** Descriptive statistics of comprehensive evaluation in Cambodia.

| | Average | Variance | Moran's I | *p* | AnselinLocal Moran's I | *p* |
|---|---|---|---|---|---|---|
| AEE | 0.5057 | 0.1584 | 0.4367 | 0.0001 | 0.0000 | 0.0119 |
| RE | 0.2704 | 0.1449 | 0.2278 | 0.0149 | 0.0000 | 0.0009 |
| AP | 0.3562 | 0.1169 | | | | |
| D | 0.5134 | 0.0824 | | | | |
| T | 0.3532 | 0.1015 | | | | |
| C | 0.7951 | 0.2238 | | | | |

Note: D, T and C respectively represent coupling coordination level, overall development level and coupling level.

There was a fault-like gap in Cambodia's road system. Road construction in the vast majority of areas was at an extremely low level. Only in the bordering areas of the plains near other countries, such as the capital city, and the coastal cities with good locations, were well built to some extent. In Phnom Penh, the road length per square kilometer was 980 m, and in Koh Kong it was 42 m per square kilometer. When conducting this research, we drove at a constant speed of 100 km/h at an intersection in Phnom Penh, the average time for us to reach the next intersection was 1.9 h. A trip like this in Koh Kong

takes 4 h. Systematic road construction is often closely related to regional development. The road system showed significant spatial autocorrelation, almost forming a low-value agglomeration zone at the downstream vertical line of Tonle Sap Lake. Apart from that, Svay Rieng, as one of the important areas for Cambodia's cross-border communication, was also at the low-value aggregation.

In addition, the overall agricultural eco-environment performed well. As shown in Figure 3, the coastal areas and the eastern plateau and mountainous areas ranked higher, while the Tonle Sap Lake area and the plain areas ranked lower. Significant spatial autocorrelation, as shown in Table 5, formed an agglomeration of low values in the plains areas around the capital Phnom Penh and high values in Ratanakiri, as shown in Figure 4.

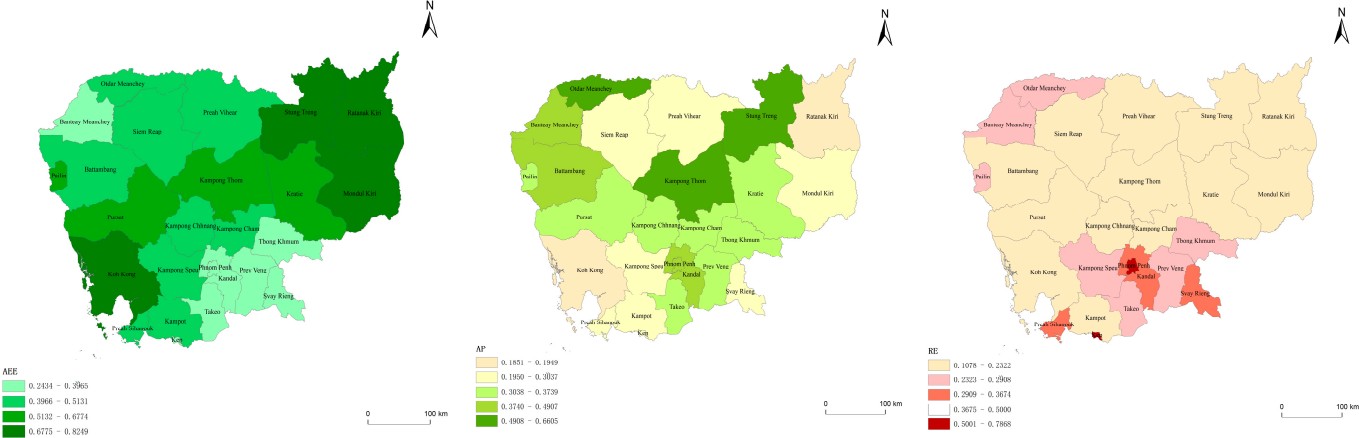

**Figure 3.** Spatial pattern of the agricultural production, road construction and agricultural eco-environment in Cambodia.

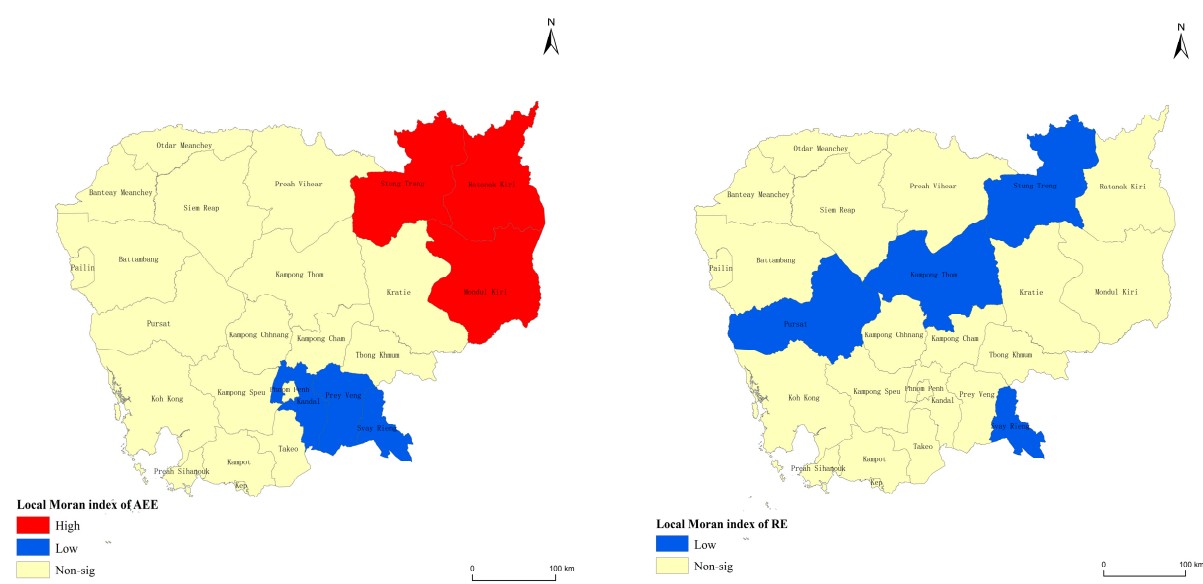

**Figure 4.** Aggregation analysis of agricultural eco-environment and road environment in Cambodia.

### 4.2. Analysis on the Coupling Characteristics of Agricultural Productio, Road Construction, and the Agricultural Eco-Environment

The overall coupling coordination degree (D) of Cambodia was at the low coordination stage (0.4–0.6), as shown in Table 5. Among the 25 subordinate provinces and municipalities, 18 provinces and municipalities were in the low coordination stage, accounting for 72%, while four provinces and municipalities were in the basic coordination stage (0.6–0.8), accounting for 16%, as shown in Figure 5.

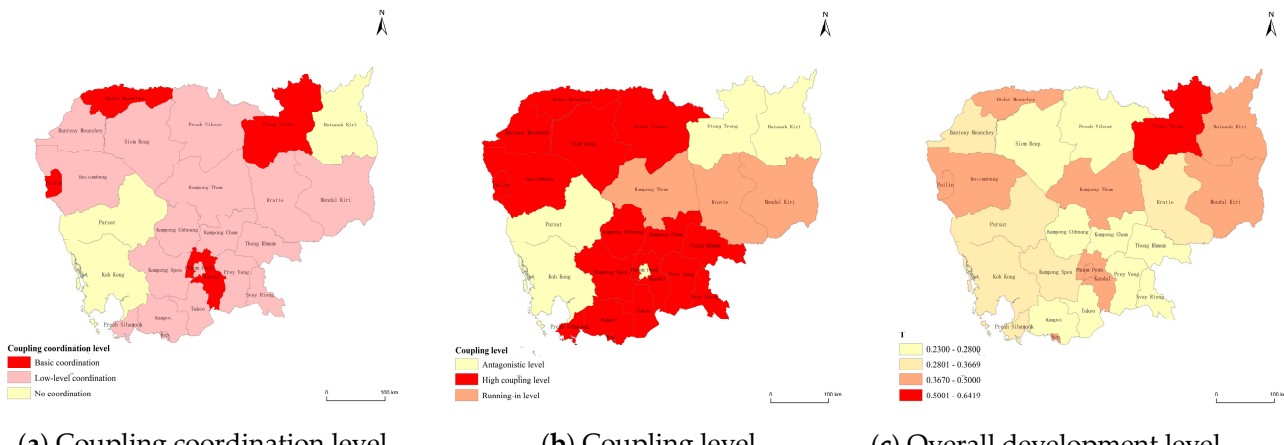

**Figure 5.** Coupling characteristics of agricultural production, road construction, and agro-ecological environment in Cambodia.

The main reason for the low coupling was the low comprehensive level of the integration (T) of agricultural production, road construction, and the agro-environment in Cambodia. What's more, in terms of the coupling level (C), the mean value was 0.7952, which indicated that there was a certain interaction between the systems. Among them, there were 20 areas with a high coupling level of grinding and above, and 17 areas with a high coupling level (0.8–1.0), accounting for 68% of the total number of areas. However, as for the comprehensive level, the average was only 0.3533. The only areas with comprehensive levels higher than 0.5 were Kampong Thom, Pailin, and Oddar Meanchey in Tonle Sap; Mondulkiri and Stung Treng in the highland mountainous area; and the capital Phnom Penh.

In terms of agricultural production, the agricultural eco-environment near the capital Phnom Penh and in the Tonle Sap Lake area of Kampong Thom, Banteay Meanchey, and Oddar Meanchey exceeded the limit of what it can bear. Thus, enhancing local agricultural production is imperative. Meanwhile, there is potential for further advanced development of the agricultural production system in the Tonle Sap Lake area and the highland mountainous areas in the east and west, but it ought to be done according to each area's unique geographical context.

As for road construction, most of the road systems in Cambodia do not have an overly negative effect on the local ecosystem. Nevertheless, in areas with a high level of economic development (Phnom Penh, Pema, Svay Rieng), roads do pose environmental risks, such as land erosion and a reduction in species diversity that could possibly lead to crises, as shown in Figure 6. What's more, lagging road development mostly occurred in Tonle Sap Lake where the highland mountains are suitable for agricultural production and where the overall assessment of agricultural production is relatively high for Cambodia. Even the road systems created by economic exchanges with other countries are unable to meet the requirements of local agricultural production. As a result, the economic value of agricultural products is not fully utilized [40]. At the same time, the development potential that roads should bring to areas with higher economic development is not fully realized, and there is a need and an opportunity to transfer the rural labor force to other labor-intensive industries, such as light manufacturing, to help these industries develop.

To sum up, as shown in Table 6, Cambodia has the following problems in the fields of agricultural production, road environment and the agro-environment: (1) potential environmental risks in the Tonle Sap Lake area and the plain areas. (2) Environmental risks of roads and the need for rural labor transfers in areas with a high level of economic development. (3) Lagging road construction in most of the main agricultural production areas. (4) Great potential but limited level of agricultural production development and utilization in some areas.

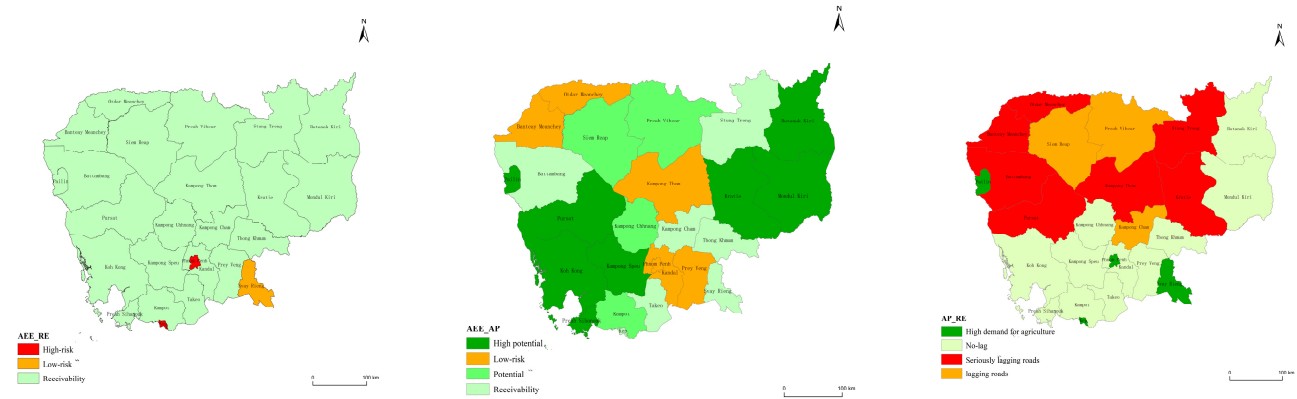

(**a**) Relationship between RE and AEE     (**b**) Relationship between AP and AEE     (**c**) Relationship between AP and RE

**Figure 6.** Relationship among agricultural production, road environment, and agricultural eco-environment in Cambodia.

**Table 6.** Relationship between agriculture, roads, and agricultural eco-environment systems in Cambodia.

| | Properties | | Region | | | |
| AEE-RE | AEE-AP | RE-AP | Plain | Tonle Sap | Costal | Plateau |
|---|---|---|---|---|---|---|
| Acceptable | Acceptable | No lag | Takeo Tboung Khmum Kampong Cham | | | |
| | | Road lag Severe road lag | | Battambang | | Stung Treng |
| Acceptable | Low risk | No lag | Kandal | | | |
| | | Severe road lag | | Banteay Meanchey Kom pong Thom Oddor Meanchey | | |
| | | Agricultural demand | Svay rieng | | | |
| Acceptable | High potential | No lag Road lag | | Kampong Chhnang Siem Reap | Kampot | Preah Vihear Kampong Speu |
| Acceptable | High potential | No lag | | Palin City | Khétt Ka Preah Sihanouk City | Môndôl Kiri |
| | | | | | | Ratanakir Kratie |
| | | Severe road lag | | Khétt Purthisat | | |
| High risk | Acceptable Low risk | High agricultural demand | Phnom Penh | | Kep | |

### 4.3. Analysis of the Interaction among Agricultural Production, Road Construction, and the Agricultural Eco-Environment

In order to adequately explore the current problems in Cambodia, the interaction between agriculture, roads, and the agricultural eco-environment was analyzed using the gray association model. Generally, a light association was considered when $0 < \gamma(X_0, X_i) \leq 0.30$, a moderate association when $0.30 < \gamma(X_0, X_i) \leq 0.60$, and a strong association when $0.60 < \gamma(X_0, X_i) \leq 1.0$ [41].

Regarding the input of agricultural production, the contribution of roads for improving the quality of agricultural labor is all-encompassing, as shown in Table 7. Currently, the density of the road network in Cambodia has the greatest impact on physical labor, the intellectual input of the labor force, and the percentage of the rural population engaged in agriculture. Road accessibility has the greatest influence on the share of the agricultural population in the greater population. With regard to the environment in Cambodia, the reduction in the agricultural population and the improvement of farmers' morality and behavior will lead to a greater stabilization of the ecological environment [42,43]. Apart from

that, fertilizers compensate for the efficiency lost due to the agricultural population shift and increase the efficiency of agricultural production [44], thus decreasing the agricultural labor time. Meanwhile, the education level of agricultural laborers could improve fertilizer utilization and reduce disturbances to the ecosystem. On the other hand, the increased rural population puts pressure on agricultural households. The increase in the population engaged in agricultural production increases the multiple cropping index and improves the utilization of arable land while contributing to an increase in the fragmentation index. In addition, the physical input of labor has the potential to increase the multiple cropping index while reducing the structural stability of the ecosystem.

**Table 7.** Interaction between the input end, road infrastructure, and agricultural eco-environment.

| Relevance | Labor | Education | Social Weight | Proportion of Agricultural Work |
|---|---|---|---|---|
| Road density | 0.6870 | 0.7314 | 0.6022 | 0.6814 |
| Grade level | 0.6600 | 0.6478 | | 0.6692 |
| Technology level | 0.6700 | 0.7026 | 0.6184 | 0.6487 |
| Accessibility | 0.6272 | | 0.6865 | 0.6365 |
| Connectivity | 0.6313 | 0.6100 | | 0.6286 |
| SHDI (ecology) | 0.7734 | 0.7115 | 0.6067 | 0.6062 |
| EVI | 0.6630 | 0.6624 | | 0.6454 |
| NPP | 0.6274 | 0.6335 | | 0.7026 |
| PD | | | 0.7118 | 0.6253 |
| Arable land per capita | 0.6836 | 0.6943 | 0.7074 | 0.7116 |
| Fertilizer use per unit area | 0.637 | 0.6424 | 0.6953 | 0.6687 |
| Multiple cropping index | 0.6485 | | 0.7117 | 0.6180 |

On the output side, the increase in per capita production value requires several aspects: the moderate use of fertilizers, an improvement in the multiple cropping index, a reduction in rising production costs caused by landscape fragmentation [45], and the improvement of road accessibility to expand the market range, as shown in Table 8. To increase the proportion of harvested agricultural products, on the one hand, a stable ecosystem is needed to reduce the loss of agricultural crops due to extreme weather. On the other hand, in terms of agricultural products, expanding their market demand via road transport, achieving their cross-regional transportation, and improving their commercialization are all necessary steps. The commercialization of agricultural products is closely related to both the road system and the agricultural eco-environment, with accessibility and fertilizer use per unit area being the most influential factors, respectively.

**Table 8.** Interactive relationship between the output, road infrastructure, and agricultural eco-environment.

| Relevance | Output Per Capita | Harvest | Commercialization |
|---|---|---|---|
| Fertilizer use per unit area | 0.6862 | | 0.7430 |
| Accessibility | 0.6896 | | 0.6972 |
| EVI | | | 0.6663 |
| Grade level | | 0.6213 | 0.6645 |
| Den | | 0.6615 | 0.6543 |
| Arable land per capita | | 0.7086 | 0.6415 |
| Multiple cropping index | 0.7388 | | 0.6366 |
| NPP | | 0.6064 | 0.6333 |
| Connectivity | | | 0.6271 |
| SHDI (ecology) | | 0.7283 | 0.6198 |
| Technology level | | 0.6343 | 0.6019 |
| PD | 0.7563 | | |

In general, as a predominantly agricultural economy, Cambodia's output side of agricultural production accounts for a relatively high proportion of agricultural production, as shown in Table 9. Various measures must be taken to improve the agricultural production capacity or increase the output. These include maintaining a per capita arable land area, ensuring ecological stability, reducing regional fragmentation, rationalizing road nodes, and providing high-quality roads as much as possible based on the local context. Focusing on the factors such as road connectivity, road accessibility, and the arable land area per capita will help to fully utilize the labor force.

**Table 9.** Interaction between agricultural production, road infrastructure, and the agricultural eco-environment.

| Relevance | Comprehensive | Output | Input |
|---|---|---|---|
| Road connectivity | 0.6365 | 0.6025 | 0.7012 |
| Arable land per capita | 0.6778 | 0.6828 | 0.6685 |
| Accessibility | | | 0.6591 |
| PD | 0.6217 | 0.6039 | 0.6556 |
| Technology level | 0.6775 | 0.6896 | 0.6546 |
| Multiple cropping index | | | 0.6493 |
| NPP | 0.6553 | 0.6595 | 0.6475 |
| Road area density | 0.6739 | 0.6936 | 0.6366 |
| SHDI (ecology) | | | 0.6306 |
| EVI | | | 0.6222 |
| Grade level | | | 0.6150 |
| Fertilizer use per unit area | | | 0.6096 |

In addition, when evaluating the environmental risks of roads, the impact of road accessibility, grade level, connectivity, technology level, and other such attributes should be the key considerations, as shown in Table 10.

**Table 10.** Interactive relationship between agricultural production, road construction and the agricultural eco-environment.

| | Structure | Self-Cleaning | Panarchy | Comprehensive |
|---|---|---|---|---|
| Accessibility | 0.6890 | 0.6368 | 0.6812 | 0.6644 |
| Grade level | 0.7123 | 0.6146 | 0.6472 | 0.6415 |
| Connectivity | | | 0.6490 | 0.6219 |
| Technology level | | 0.6631 | | 0.6131 |
| Density | | 0.6317 | | |

The length of regional roads is the primary consideration in order to expand the agricultural product market and achieve an agricultural population transfer as well as more efficient agricultural production in the main agricultural production areas where the level and quality of roads are lagging. The scale of investment in road construction is large, the pay-off time is long, and sometimes even foreign teams are required to build and operate the roads. Therefore, it is also necessary to improve the length of regional roads while rationally taking into account road planning, road quality, and other factors to achieve efficient, low-budget, and long-term use of roads. Road connectivity is one of the priorities for road supply. However, for the agricultural production development in areas with potential, there are certain differences. In the coastal and highland mountainous areas in the northeast, the ecosystem is particularly sensitive to human disturbance. Thus, developing modern agriculture, improving agricultural production from the input side, and strengthening personnel exchanges with the surrounding areas is key. Apart from this, in the Tonle Sap Lake area, it is preferable to improve agricultural production from the supply side and make full use of the local land resources to achieve solid agricultural development. For some areas, to address the potential environmental risks in agricultural production, the

customs and behavior of the local residents should be emphasized to minimize disturbances to the ecosystem and improve its stability. Additionally, laws should be formulated in conjunction with publicity initiatives to mitigate agricultural landscape fragmentation.

High economically developed areas have road environmental risks. A reasonable assessment based on the attributes of the roads should be conducted, as well as a subsequent hedging of environmental risks and corresponding ex-ante crisis management. A cautious attitude in road planning and a research-based selection of road nodes and grades is also needed. In addition, connections with neighboring areas should be strengthened to enhance the ease of public mobility. Though there are still some areas with sound coordinated development, their overall level of development is not high. Therefore, further related work should be carried out, particularly taking the local context into account.

## 5. Discussion and Conclusions

### 5.1. Discussion

This study evaluated agricultural production, regional road construction, and the agricultural ecological environment, putting forward a theoretical framework through building a comprehensive evaluation index system. By taking Cambodia as an example, it further applied the coupling coordination degree model and gray correlation model to explore the interactive relationship among agricultural production, road construction, and the agricultural ecological environment. Based on the social–ecological resilience theory, the study attempted to seek the most suitable development paths for different regions under the premise of ensuring environmental sustainability. Focusing on a human–land nexus, this study paid particular attention to the coordinated development of the agricultural eco-environment, agricultural production, and road infrastructure. To understand the coupling effects between them, it was of utmost importance to select the proper indicators to measure the agricultural eco-environment effected by agricultural development. In addition, how to effectively utilize the landscape index and other environmental indicators in the selection of the indicators is an issue worthy of further discussion. This study classified the indicators in the previous related studies, combined the relevant ecological knowledge, and then replaced them with landscape indicators. Nevertheless, the ecosystems had natural externalities, and thus the outcome of their subsystems' interaction with a specific human domain was often disturbed by the behavior of other domains, i.e., humanity's negative effect on the environment is multi-faceted, which is likely to result in an even harsher reality.

The study revealed that in Phnom Penh, Kep, and Svay Rieng, due to the relatively high level of economic development, road construction faced relatively higher ecological and environmental risks such as soil erosion, excessive heavy metals, and a sharp decline in species diversity. As agricultural production developed to a certain level, there was a need for rural labor shifting to other labor-intensive industries such as the light industry and manufacturing [24]. In the Tonle Sap Lake and the plains region, agricultural development was still at a relatively lower level of development due to road construction lagging far behind, though there was a huge potential for agricultural production. In the plain areas, the growing population brought tension between food security, fertilizer abuse, and deforestation, which intensified the disturbances to the agricultural ecological environment.

High-quality roads should be provided as much as possible according to the local population and regional area, and road connectivity should be made a road supply priority. Proper research for future infrastructure planning such as road nodes and road grades should be considered in order to enhance its role in promoting economic development [46]. For the coastal and highland mountainous areas in the northeast, there is a tendency to improve agricultural production from the input side [45]. For the Tonle Sap Lake region, it is preferable to improve the agricultural production level from the supply side. In general, the work should be done gradually to avoid detrimental costs caused by overly adventurous goals. The potential environmental risks can be addressed through organized short-term training for residents, which includes improving the efficiency of fertilizer use and crop

management [29]. In addition, environmental risk response plans can be formulated to improve the management level of responding to ecological and environmental risks. Laws can be developed and coupled with publicity measures to mitigate agricultural landscape fragmentation.

*5.2. Conclusions*

Food security has always been a key component of sustainable development for all countries. Transportation and logistics are not only key issues for agriculture profitability, but for the effective allocation of human resources. In Cambodia, better road development can improve the accessibility of rural markets, promote non-agricultural employment, increase the income of rural residents, and ensure food security [46,47]. Meanwhile, road infrastructure is the most urgently needed infrastructure for smallholders in rural areas. Improved road construction is conducive to accelerating the transformation of smallholder agriculture into intensive agriculture, including the possibility of diversifying non-agricultural income [48]. However, road construction and the expansion of farmland leads to a series of ecological and environmental problems, such as disturbed bird breeding in forests, excessive use of fertilizer on farmland, the reduction in forest areas and the reduction in biodiversity [49–51]. When agricultural development, road construction, and ecological security are all placed on the policy agenda, balancing the development needs in different regions becomes the priority for policymakers.

Over the past decades, rapid economic development in many countries has been accompanied by an increase in agricultural systems output and a shift in agricultural populations. This process, which denotes the separation of time and space between generations of families, is challenging for those involved, but it is also inevitable. Ecological agriculture is a reflection of extreme modernization, and such a reflection is more common in relatively developed countries. The products of industrialization, such as large-scale production, machinery, and chemical fertilizers, are not only replacing the traditional equivalents and shifting the agricultural population but also interfering with the ecosystem. However, most developing countries consider agriculture as a means for survival rather than a bearer of risks to the agricultural ecosystem. This research outcomes can also be applied to other developing countries that seek to improve their integration of agriculture, roads, and the agro-environment.

Numerous studies on environmental and regional development have been conducted, among which landscape ecology is one of the current popular research topics. In the future, various aspects deserve further investigation. First, more appropriate indicators should be considered based on the local context, including culture, climate, governance type, and other such factors. For example, indicators can be further analyzed based on regional agricultural development, such as pesticide usage per hectare, rainfall level, and financial expenditures in agriculture. Second, the indicators can also be further applied through principal component analyses according to the actual situation of the study area. Finally, conducting a continuous, dynamic study of a particular country would allow for greater investigation of the influencing factors that play an important role in regional development.

**Author Contributions:** Conceptualization, L.W.; methodology, W.D.; software and data curation, W.D.; formal analysis, L.W. and W.D.; investigation, L.W. and W.D.; writing—original draft preparation, L.W. and W.D.; writing—review and editing, Y.C.; supervision, L.W.; project administration, Y.C. All authors have read and agreed to the published version of the manuscript.

**Funding:** This research was supported by Chongqing University, funded by General Research and Development Project of National Higher Education Institution (No. 2020CDJSK01PY18), and the National Social Science Fund of China (No.17CGJ007).

**Institutional Review Board Statement:** Not applicable.

**Data Availability Statement:** The data that are presented in this study are available from the corresponding author upon request. Relevant links to publicly archived datasets analyzed during the study are as follows: (1) http://www.gadm.org/, accessed on 15 March 2022. (2) https://www.

openhistoricalmap.org/, accessed on 15 March 2022. (3) http://www.nis.gov.kh/, accessed on 15 March 2022. (4) https://www.fao.org/, accessed on 15 March 2022. (5) https://lpdaac.usgs.gov//, accessed on 22 March 2022. (6) http://www.globallandcover.com/, accessed on 9 April 2022.

**Acknowledgments:** We thank Jeffrey Sayer, Shenggen Fan, Chris Margules, Rebecca Riggs and Madeleine King for their suggestions and language editing for this manuscript.

**Conflicts of Interest:** The authors declare no conflict of interest, and the funders had no role throughout the study.

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
