# Peer review of "Study on the Coupling Effect of Agricultural Production, Road Construction, and Ecology: The Case for Cambodia"

_agriculture, doi:10.3390/agriculture13040780_

Round 1

Reviewer 1 Report

Based on sustainable development theory and ecological-social resilience theory, the author established a comprehensive evaluation index system, and used the coupled coordination model and grey relation analysis model to explore the interaction among agricultural production, regional road construction and agro-ecological environment in Cambodia. This is of great significance for improving regional development in developing countries fallen into a stalemate between modernization and development. Here are some suggestions for your reference.

1.    Can you tell us a little bit about Cambodia behind the data? The paper as a whole is data-driven and lacks qualitative analysis.

2.    The innovation point seems to be not clearly described, please condense the language to make it more accurate.

3.    Is there a coupling or a causal relationship among agricultural production, regional road construction, and agricultural eco-environment? Please explain the relationship between the methods.

4.    Please indicate the source of the indicators selected in Table 1.

5.    The first paragraph of the discussion section (lines 502-507) seems unnecessary and is proposed to be deleted. As of now, there is still room for further depth in the discussion section, so please continue to deepen it in conjunction with the results section.

6.    It is recommended that the figure be revised to make it more beautiful and meet publication requirements.

Reviewer 2 Report

Food security has always been the key factor of the sustainable development of any country. Balance between agricultural technology and environmental protection is a very important feature of sustainable territory development. Transportation and logistics are key issues for agriculture profitability.

The study could benefit if the authors had considered the location of agricultural land and farmstead or other facilities, i.e. issues like far-away placement, strip farming, and its effects on the profitability of agriculture.

Reviewer 3 Report

In the abstract, the striking findings of this study is conspicuously missing. In line 20 and other parts of the main text, authors are advised to change "construction" to "development". I will recommendation rewriting the abstract. 

From the introduction, I feel what you meant by "agricultural modernization" is "agricultural development". Can you replace it? What do you mean by "hot lands", "ASEAN"? Give the meanings at first mention. 

In line 154-156, support with reference(s). There was no references in the overview and days source section (line 154-173). Give the meaning of GADM (line 178) at first mention.

Support equation 1 with relevant reference(s). 

In line 228, you're expected to have defined agro-environment in the earlier paragraphs in the introduction.

In section 3.2, you're expected to mention some previous empirical studies with references (line 247-250). Add source to all the Tables added in this study. 

In Table 1-2, I feel authors have not presented a detailed explanation for better understanding of your readers. Even in other Tables, you are expected to give concise explanation of these Tables. I found it difficult to comprehend what the data were pointing at. Your data source was vague and when this happens, it will be difficult to understand your results and discussion sections.

In Table 4, how did you arrive at the wight assigned to each variable? How did inform your choice of the variables? 

In the discussion, no references was cited to either support or refute the findings of this study. This is unacceptable. You're expected to concisely compare the findings in this study with those of the previous empirical studies in literature to either support or refute the findings.

Round 2

Reviewer 3 Report

Thank you for sending the revised version of this manuscript. The discussion presented in this study is inadequate  is still grossly for the results presented. Kindly give detail discussion of the findings of this study. Thank you.

Author Response

 We have adressed the reviewer's comments and revised the section of discussion.